# Novel Treatments for Obesity: Implications for Cancer Prevention and Treatment

**DOI:** 10.3390/nu15173737

**Published:** 2023-08-25

**Authors:** Carla Micaela Cuttica, Irene Maria Briata, Andrea DeCensi

**Affiliations:** 1Endocrine Unit, E.O. Ospedali Galliera, 16128 Genoa, Italy; 2Division of Medical Oncology, E.O. Ospedali Galliera, 16128 Genoa, Italy; irene.maria.briata@galliera.it (I.M.B.); andrea.decensi@galliera.it (A.D.); 3Wolfson Institute of Population Health, Barts and the London School of Medicine and Dentistry, Queen Mary University of London, London E1 2AD, UK

**Keywords:** obesity, neoplasms, anti-obesity agents, glucagon-like peptide 1, liraglutide, weight loss

## Abstract

It is now established that obesity is related to a higher incidence of cancer during a lifespan. The effective treatment of obesity opens up new perspectives in the treatment of a relevant modifiable cancer risk factor. The present narrative review summarizes the correlations between weight loss in obesity and cancer. The current knowledge between obesity treatment and cancer was explored, highlighting the greatest potential for its use in the treatment of cancer in the clinical setting. Evidence for the effects of obesity therapy on proliferation, apoptosis, and response to chemotherapy is summarized. While more studies, including large, long-term clinical trials, are needed to adequately evaluate the relationship and durability between anti-obesity treatment and cancer, collaboration between oncologists and obesity treatment experts is increasingly important.

## 1. Introduction

The obesity pandemic is on the rise around the world; by 2030, it is expected that 1 in 5 women and 1 in 7 men will live with obesity, equal to over 1 billion people globally, producing a considerable impact on the number of years of life lost due to disease and premature death [1].

Obesity is a chronic, relapsing and multifactorial disease linked to excessive adiposity that predisposes individuals to other non-communicable diseases (NCDs) and medical complications, such as type 2 diabetes (T2DM), cardiovascular disease, chronic kidney disease, gallbladder disease, non-alcoholic fatty liver disease, gout, obstructive sleep apnea, osteoarthritis and cancer [2]. It is now known that obesity can increase the risk of different types of cancer, including cancers of the breast in postmenopausal women, colorectum, endometrium, kidney, liver, gallbladder, ovary, pancreas, gastric cardia, esophagus, thyroid, multiple myeloma and meningioma, and may worsen the survival of patients with cancer, especially for breast, bladder, colorectal, prostate and liver cancers. Obesity is estimated to overtake smoking as the main risk factor for cancer in many countries in the coming decades, making obesity treatment a new challenge in cancer control strategies [3,4]. Obesity often starts early in life and may influence the likelihood of developing one of the related NCDs in adulthood. Furer et al., in a large population-based cohort of 2.3 million Israeli adolescents, found that a high body mass index (BMI) in adolescence is positively associated with an increased risk of mortality after 10 years for all types of cancer in both sexes [5]. In addition, other authors have found an association between shorter survival and cumulative exposure to a higher BMI during early to mid-adulthood in patients with breast and colorectal cancer, irrespective of the presence of cardiometabolic disease prior to cancer diagnosis, pointing out how effective early life intervention strategies on BMI can become an important target of preventive intervention [6].

It is important to point out how reductive it is to talk about obesity based only on BMI. It is the excess of adiposity that gives rise to complications that compromise health and confer increased morbidity and mortality; therefore, the preferable term today is adiposity-based chronic disease (ABCD), which motivates the importance of excessive distribution of fat [7]. Adipose tissue distribution is of increasing interest due to its role in cancer development, as shown in a recent study of postmenopausal women with breast cancer with a normal BMI but high body fat evidenced with dual-energy X-ray absorptiometry (DXA) [8]. Adipose tissue can be classified into several different subtypes by function and anatomical location: white, brown and beige adipose tissue. Brown and beige adipose tissue deal with thermoregulation, whereas white adipose tissue is responsible for the energy storage of lipids (triacylglycerides) and is further divided into different compartments: subcutaneous adipose tissue and visceral adipose tissue. Visceral adipose tissue is more lipolytically active, contributing to the increase in free fatty acids and thus resulting in insulin resistance and to a more pro-cancer secretome than subcutaneous adipose tissue [9]. The accumulation of visceral adipose tissue in the body, in fact, contributes to an increase in tissue inflammation (through the infiltration of immune cells and the formation of crown-like structures by the macrophages that surround dead or dying adipocytes) and to an altered secretion of inflammatory factors that, in turn, are associated with the development of insulin resistance, metabolic syndrome and increased cancer risk and worse prognosis.

Furthermore, the importance of adipose tissue in tumorigenesis is supported by the finding that cold exposure produces a browning in adipose tissue that increases thermogenesis and glucose uptake while reducing glucose uptake, tumor growth, proliferation and hypoxia in cancer cells, making adipose tissue a possible target of cancer therapy by mediating tumor suppression through cold-impaired metabolism [10].

BMI alone does not distinguish between lean mass and fat mass, but it is inexpensive and widely used as is waist circumference, an indirect measure of abdominal obesity related to abdominal visceral fat. More accurate assessments of body fat distribution (e.g., bioelectrical impedance, DXA and magnetic resonance imaging) are expensive and limited in use in epidemiological studies [9].

Three mechanisms are mainly used to explain the relationship between obesity and cancer: sex hormone impaired metabolism, impaired insulin signaling and an excess of pro-inflammatory cytokines [3]. Weight gain in adulthood and increased body fat are known to increase the risk of endometrial cancer and the risk of hormone receptor-positive breast cancer in postmenopausal women in whom adipose tissue is the principal site for estrogen synthesis, making estrogen production a major contributor in women with obesity. Obesity is often characterized by hyperinsulinemia and insulin resistance, which are capable of stimulating the growth of cancer cells directly through insulin receptors and indirectly through insulin-like growth factor (IGF) receptors by inhibiting apoptosis and promoting cell growth, motility and invasion. Hyperinsulinemia also reduces sex hormone binding globulin levels (increasing free circulating estrogens that can promote estrogen-dependent tumors), reduces IGF-binding protein by increasing free IGF and increases the production of pro-inflammatory cytokines and adipokines, thus supporting the chronic low-grade inflammation implicated in the development of cancer, as illustrated elsewhere [3,9]. As far as adipokines are concerned, it is known that, in obesity, adiponectin decreases, and leptin increases. Leptin appears to be a pro-oncogenic factor: it is able to promote systemic inflammation, cell proliferation, angiogenesis and metastasis, also inhibiting apoptosis, immune surveillance and cell death; however, adiponectin has opposite effects. The imbalance between leptin and adiponectin plays an important role in the obesity–cancer relationship [9,11].

The recent novel adipokine circulating miRNAs, acting as epigenetic regulators, are capable of influencing adipocyte differentiation, white adipose tissue browning, lipid and glycemic homeostasis and insulin resistance with an essential role in obesity-associated inflammation, metabolic syndrome and predisposition to cancer. Moreover, miRNAs, extensively produced by adipose tissue in both paracrine and endocrine signaling, are able to participate in oncogenic and tumor suppressor molecules, modulating the cancer microenvironment, settlement, progression and invasion [12]. Gut hormone dysregulation is also involved in the pathogenesis of obesity, and emerging data hypothesize its implication in cancer development [3,13]. Gut hormones are secreted by entero-endocrine cells in the gastrointestinal tract; they react to the deficit or arrival of nutrients by acting as regulators of eating habits/appetite/satiety and regulators of energy homeostasis; they are also able to regulate insulin secretion. The dysregulation of gut hormones present in obesity is implicated in excessive nutrient intake, impaired insulin signaling leading to hyperinsulinemia and T2DM and chronic low-grade inflammation arising from adipose tissue, which are all mechanisms involved in the development of cancer [13].

The treatment of obesity, therefore, appears essential to reduce the risk of cancer.

## 2. Weight Loss and Cancer Risk and Prognosis

Intentional weight loss (including diet/exercise/surgery), especially if greater than 10%, is able to reverse the proinflammatory state linked to obesity, e.g., reducing the levels of C-reactive protein, tumor necrosis factor-α, interleukin-6 and the leptin-to-adiponectin ratio. Data are limited because adipokines and cytokines are difficult to measure due to their low concentrations; however, although further investigation is needed in this area, many studies suggest that weight loss among overweight or obese people is helpful in reducing cancer risk [3,9,14,15]. Moreover, it is known that subjects with obesity/ABCD at the time of cancer diagnosis have worse outcomes and poorer survival outcomes [3,16]. In breast cancer, obesity seems to be linked to a heightened risk of recurrence and overall mortality with an estimated risk that ranges from 35% to 40% [16,17]. Importantly, this correlation persists regardless of menopausal status, hormone receptor status or specific cancer subtypes [17,18]. Also, being overweight or obese appears to be linked with the occurrence of distant and delayed recurrences in patients with breast cancer [17].

The “obesity paradox” debate that has occurred between the association of high body weight and cancer survival benefits and increased tolerance during some anticancer therapies may be explained by the association of obesity with subtypes of less aggressive tumors and other confounding factors, such as smoking and body composition [16]. Intervention studies predominantly conducted in breast, endometrial and prostate cancer trials showed positive effects of intentional weight loss based on diet composition, caloric intake and amount and nature of physical activity [19,20,21,22]. It is also known that bariatric surgery, achieving significant and long-term weight loss, is effective in reducing the incidence of obesity-associated cancer and the cancer-related mortality [23].

Weight gain following a breast cancer diagnosis also seems to slightly elevate the risk of overall mortality, and the harmful impact was most pronounced when weight gain exceeded 10% [17]. Therefore, the evaluation of BMI at the time of diagnosis may be the most significant predictor of breast cancer prognosis [17], and strategies aimed to control that increase are crucial. A recent randomized trial provided evidence for the favorable impact of a weight loss intervention through diet, physical activity and behavioral change on outcomes among women with breast cancer [24].

In individuals with prostate cancer, an increase of 5 kg/m^2^ in BMI was found to correspond to a 9% higher risk of prostate cancer-specific mortality and a 3% higher risk of all-cause mortality [25]. Additionally, there is a moderate–consistent association between obesity and biochemical recurrence following radical prostatectomy [26]. Despite the role of obesity in prostate cancer, only a few prostate cancer guidelines recommend the adoption of a healthy lifestyle, and only 7.2% provide advice on reaching or maintaining a healthy weight [22].

It should be emphasized that weight management strategies in overweight and obese cancer survivors may also play an important role in preventing non-cancer deaths. Diabetic patients with cancer have reduced overall survival compared with non-diabetics, in particular, related to an increased risk of non-cancer deaths, primarily cardiovascular, which may be further increased by some cancer treatments, and weight loss programs have been shown to be effective in the remission of diabetes [27].

## 3. Metformin and Cancer

Many studies have documented an association between metformin, a biguanide for the treatment of T2DM, and reduced cancer incidence and mortality [28].

Among high-risk patients with operable breast cancer without diabetes, adding metformin versus a placebo to standard cancer treatment did not significantly improve invasive disease-free survival in a large phase III trial [29]. However, patients with ERBB2+ breast cancer in the metformin group vs. the placebo group had longer invasive disease–free survival and overall survival [29]. Moreover, in vitro studies support the efficacy of metformin in cancer therapy and prevention, and a review of epidemiological studies on metformin treatment showed a positive trend for benefit [30,31]. In a randomized placebo controlled presurgical trial, metformin was able to reduce breast cancer cell proliferation in women with insulin-resistance, suggesting a potential effect of the drug in this subgroup of patients [32,33].

Multiple mechanisms are involved: (1) metformin indirectly reduces tumor proliferation via insulin-lowering activity in subjects with hyperinsulinemia, involving the insulin/IGF-1 pathway, which is known to contribute to cell growth and proliferation by activating both phosphatidylinositol-4,5-bisphosphate 3-kinase (PI3K)/Akt/mTOR and the Ras/Raf/mitogen-activated protein kinase (MAPK) pathways; (2) metformin also indirectly affects cancer cells by modulating the immune response via an anti-inflammatory effect; (3) metformin acts directly on respiratory Complex I of the electron transport chain in the mitochondria of pre-neoplastic and neoplastic cells, reducing the energy consumption of cells; (4) metformin is able to inhibit the activation of matrix metalloproteinase-9, blocking the invasion of cancer cells, and to activate growth suppressants through phosphorylation of the retinoblastoma protein [34,35]. Metformin, acting through suppression of the mitochondrial electron transport chain complex I, reduces mitochondrial ATP, which increases the adenosine monophosphate (AMP): ATP ratio; as a result, AMP-activated protein kinase is activated, which inhibits hepatic lipogenesis and gluconeogenesis [36]. Metformin-induced activation of the AMPK pathway is able to reduce the tumor-promoting activity of insulin and inhibit mTOR, which is closely related to tumor cell proliferation, making metformin an intriguing molecule in cancer therapy [37,38]. Due to modest weight-loss effects, metformin is not approved as a weight-loss agent, but it is often used as an off-label drug in obese patients with prediabetes, insulin resistance, metabolic syndrome and polycystic ovary syndrome, especially in the absence of lifestyle changes or when other anti-obesity drugs cannot be used [39]. The Diabetes Prevention Study has shown that metformin is able to reduce the incidence of diabetes and produce a slight but persistent reduction in body weight (approximately 2–3 kg), BMI and waist circumference in subjects with a high risk of T2DM [40]. The metformin-induced weight change is not attributed to an increase in energy expenditure but rather to a reduction in caloric intake via appetite suppression. One hypothesis is that metformin, by suppressing the mitochondrial complex I of the electron transport chain, reduces mitochondrial production of ATP and diverts glucose to anaerobic respiration, inducing mild metabolic acidosis by the production of lactate (particularly in the postprandial period) and thus inducing appetite suppression [41]. Metformin crosses the blood–brain barrier, acting in areas associated with food–reward relationships and acting in the hypothalamus by reducing orexigenic peptides, neuropeptide Y and agouti-related protein, possibly by increasing the signal transducer and activator of transcription 3 (STAT3) that is identified as a key mediator of feeding [42]. Metformin is also capable of acting on the suppression of appetite, promoting, through a direct local gastro-intestinal action, the secretion of the anorexic intestinal incretin hormone glucagon-like peptide 1 (GLP-1), and also through the inhibition of dipeptidyl peptidase-IV (DPP-IV), an enzyme that degrades GLP-1 [43,44].

## 4. Obesity Therapy and Cancer Treatment and Prevention

Novel specific therapeutic strategies are currently present to address the global obesity epidemic and its comorbidities, and several promising drug targets are under study. The medications currently approved in Europe for the treatment of obesity are orlistat, naltrexone/bupropion, liraglutide and semaglutide and, in the United States, short-term phentermine and a combination of phentermine/topiramate (in 2020, the US Food and Drug Administration ordered the withdrawal of lorcaserin from the market since a drug safety clinical trial showed an increased risk of cancer [45,46].

Orlistat (tetrahydrolipstatin, a synthetic derivative of lipstatin produced by Streptomyces toxytricini) is a pancreatic and gastric lipase inhibitor, which reduces the absorption of dietary fat, developed as an anti-obesity drug. Orlistat inhibiting fatty acid synthase, a lipogenic enzyme that catalyzes fatty acid synthesis, has been shown to decrease proliferation and increase apoptosis in various tumor cell lines [47,48,49,50,51].

There are currently very few studies on the effects of the combination of naltrexone/bupropion, short-term phentermine and phentermine/topiramate focused on cancer effects in obesity. In an 8-week intervention study of the effects of the Mediterranean diet and naltrexone/bupropion therapy, a fixed-dose combination product containing naltrexone, an opioid antagonist, and bupropion, an aminoketone antidepressant, in breast cancer survivors overweight or obese did not show superior effects when compared to the Mediterranean diet alone [52,53].

Liraglutide and semaglutide are two GLP-1 receptor agonists (GLP-1RAs) formerly known as antidiabetic drugs for the treatment of T2DM and currently approved, in higher dosages, for chronic weight management in addition to diet and physical activity in patients with obesity (BMI ≥ 30 kg/m^2^) or who are overweight (BMI ≥ 27 kg/m^2^) with at least one weight-related comorbidity (such as hypertension, T2DM, dyslipidemia): liraglutide 3 mg subcutaneous daily (approved for obesity in adults and adolescents 12 years and older) and semaglutide 2.4 mg subcutaneous weekly [54,55,56,57].

GLP-1 is an incretin hormone secreted by gastrointestinal L cells in response to nutrients in the lumen; it accounts for up to 70% of insulin secretion in response to nutrient intake, and it also lowers glucagone secretion while minimizing hypoglycemia. GLP-1 is capable of suppressing appetite and delaying gastric emptying, both of which are responsible for its slimming effects [58]. GLP-1 inhibits food intake, mediating appetite suppression by directly stimulating POMC/CART neurons and indirectly inhibiting neurotransmission in neurons expressing neuropeptide Y and agouti-related peptide via GABA-dependent signaling [59]. GLP-1 half-life is very short (less than 2 min), and it is rapidly degraded by enzymes (DPP-IV and neutral endopeptidase -NEP). Semaglutide and liraglutide are modified long-acting analogues of native GLP-1 (Aib 8, Arg 34 and Arg 34 substitutions, respectively). With the addition of an albumin-binding C16 fatty acid side chain, the half-life of liraglutide is 13–15 h. The half-life of Semaglutide is 165 h, resulting from an amino acid replacement (preventing DPP-IV degradation) and the addition of a C18 fatty diacid. Additionally, semaglutide has positive effects on hedonic aspects and reward-related behaviors of food intake (e.g., reducing cravings and lowering altered food preference) [45,56,60]. However, they are costly and may have adverse effects in some individuals, the most common mainly at the beginning of the treatment cycle and consisting of gastrointestinal disturbances (including nausea, diarrhea, vomiting, constipation), which are mostly mild, transient and dose-dependent but may lead to discontinuation of treatment in some patients [58].

In addition to diet and physical activity, liraglutide is capable of producing at least 5–10% weight loss compared to the placebo, and semaglutide is capable of producing a mean reduction of 14.9% in body weight from baseline, semaglutide being more effective than liraglutide in weight loss (10% or more: 70.9% of participants vs. 25.6%, 15% or more: 55.6% vs. 12.0% and 20% or more: 38.5% vs. 6.0% respectively; all *p* < 0.001) [61,62,63].

GLP1-receptors (GLP-1Rs) are widely distributed in the body, and in addition to the effects mentioned above, they have multiple biological effects: cardioprotective, neuroprotective, reduction in neuroinflammation and fat deposition, appetite suppression and gastric emptying delay [64].

Notably, liraglutide compared to the placebo in patients at high risk of cardiovascular disease and without T2DM was able to significantly reduce visceral adipose tissue (−12.5% vs. placebo −1.6%) and levels of CRP (−38%) [65].

Semaglutide, on the other hand, appears to have positive effects on skeletal muscle; can significantly reduce body weight and intramuscular fat accumulation; promote muscle protein synthesis; increase the relative proportion of skeletal muscle and improve muscle function in obese mice by decreasing the levels of triglycerides, serum cholesterol, low-density lipoprotein, high-density lipoprotein, TNF-α, IL-6, IL-1β and fasting insulin resistance index (HOMA-IR) and significantly improving type I/type II muscle fiber ratio, total muscle fiber area, muscle fiber density, sarcomere length and mitochondrial number and area [66].

GLP1-RAs (receptor agonists) also show interesting perspectives on tumor development and prognosis. Meta-analyses and systematic review of clinical trials demonstrated that, in obese patients (with or without T2DM), GLP-1RAs do not increase the risk of breast cancer, pancreatic cancer, thyroid cancer and malignancies in general [67,68,69], although some authors underscore concerns about potential tumor-related adverse effects when combining GLP-1RAs with DPP-IV inhibitors [70].

Furthermore, there is still controversy regarding the possible association between GLP1-RAs and thyroid cancer risk: some authors showed correlations while others showed no correlation, also pointing out potential surveillance bias [71,72,73]. However, although there is no clear evidence of the development of c-cell malignancies, it should be noted that GLP-1RAs are currently not recommended in patients with family or personal history of medullary thyroid cancer, as well as multiple endocrine neoplasia syndrome type 2 [74,75]. Further data are expected in the coming years from targeted clinical trials [76].

Randomized clinical trials reported the incidence of malignant neoplasms in overweight or obese adults as 0.7% (3/407; basal cell carcinoma, breast cancer and papillary thyroid cancer) and 1.1% (6/535; breast neoplasms, endometrial adenocarcinoma, marginal zone lymphoma and melanoma) in the semaglutide group and 0.5% (1/204; invasive lobular breast carcinoma) and 0.4% (1/268; metastatic lung cancer) in the control group, respectively [77,78]. In a recent randomized clinical trial comparing liraglutide to semaglutide, malignancies occurred in 2.4% with semaglutide (3/126; basal cell carcinoma, clear cell renal cell carcinoma and invasive ductal breast carcinoma) in 2.4% with liraglutide (3/127; basal cell carcinoma, invasive ductal BC and invasive lobular breast cancer) and 1.2% with placebo (1/85; invasive ductal breast cancer) [63].

### 4.1. GLP-1 Receptor Agonists and Breast Cancer

Experimental studies investigated the effects of GLP-1 RAs on breast cancer. GLP-1R is expressed in human breast cancer tissue and cell lines. By inhibiting nuclear translocation of NF-κB and target gene expression, GLP-1RA exenadin-4 dose-dependently reduced the growth of breast cancer cell lines in vitro as well as in vivo in transplanted athymic nude mice. In contrast, a DPP-IV inhibitor (linagliptin) did not affect breast cancer cell proliferation, suggesting that GLP-1 might attenuate cell proliferation through GLP-1R activation [79].

Liraglutide in breast cancer cell lines is able to modulate epigenetics, reducing cell viability, migration, DNA methyltransferase activity, decreasing the DNA methylation profile for *CDH1*, *ESR1* and *ADAM33* gene promoter regions and, consequently, increasing their expression. Furthermore, liraglutide and the combination treatment of liraglutide and paclitaxel or methotrexate was shown to be effective in reducing tumor growth in in vivo studies by modulating *CDH1* and *ADAM33* gene expression in mice, suggesting liraglutide as a possible treatment for breast cancer [80].

Liraglutide, in a dose-dependent manner, was able to reduce proliferation and increase apoptosis in human breast cancer cell line MCF 7 cells compared with the control group by inhibiting the expression of miRNA-27a, which subsequently increases the expression of adenosine monophosphate-activated protein kinase (AMPKα2 protein), the authors suggest the experimental basis for clinical breast cancer treatment strategies [81].

Liraglutide has been shown to have anti-proliferative effects on MCF-7 human breast cancer cells cultured in obese adipose tissue-derived stem cells-conditioned medium (ADSCs-CM -the multipotent mesenchymal lineage implicated in breast cancer development, invasion and metastasis, by the secretion of cytokines, adipokines, interleukins, tumor necrosis factor-alpha and growth factors that promote growth, migration and invasion of breast cancer cells). The drug was able to decrease the levels of inflammatory mediators, suppress the pro-proliferative effects of leptin and enhance the anti-proliferative effects of adiponectin in ADSCs-CMs. Therefore, the authors suggest that liraglutide could mitigate breast cancer cell growth in obese subjects [82].

However, liraglutide at higher concentrations in an approximate toxicological context in human triple negative breast cancer cells (MDA-MB-231 and MDA-MB-468) and in transplanted tumors was shown to be able to promote progression through the signal NOX4/ROS/VEGF pathway after GLP-1R activation, suggesting attention to dosages and breast cancer phenotype [83].

In the SCALE trials evaluating the effect of liraglutide in the treatment of obesity, a non-significant higher incidence of breast cancer was observed in the treatment group (15 vs. 3 events; incidence, 4.36 vs. 1.80 events for 1000 person-years) in women who had greater weight loss in the first year of treatment and thus partially attributed to detection bias due to facilitation of mass detection during breast exams [61]. In the UK Clinical Practice Research Datalink national database, the use of GLP-1RAs was not associated with an increased risk of breast cancer over an average follow-up of 3.5 years [84]. A recent systematic review and meta-analysis assessing the incidence of breast cancer in subjects treated with GLP-1RAs found no increased risk [67].

### 4.2. GLP-1 Receptor Agonists and Prostate Cancer

Interesting perspectives between GLP1-RAs use and prostate cancer are emerging. The randomized, double-blind, controlled LEADER study (Liraglutide Effect and Action in Diabetes: Evaluation of Cardiovascular Outcome Results) in a secondary outcome analysis showed a lower incidence of prostate cancer in the liraglutide group (*n* = 26) compared to the placebo (*n* = 47) with HR 0.54 (95% CI: 0.34–0.88) [85]. Lu et al., in a large UK Clinical Practice Research Datalink patient cohort (2007 to 2019), found that GLP-1RAs were associated with a reduced risk of prostate cancer compared with sulfonylurea use (rates incidence = 156.4 vs. 232.0 per 100,000 person-years, respectively, HR = 0.65, 95% CI = 0.43, 0.99) [86].

Interestingly, GLP-1Rs are expressed in human prostate cancer tissue obtained by radical prostatectomy from patients with nondiabetic prostate cancer; they are co-located with P504S/α-methylacyl-CoA racemase, a cytoplasmic protein identified as a sensitive and specific positive marker for prostate carcinoma. GLP1-RA Exenedin-4 (Ex-4), in a dose-dependent manner, significantly reduced prostate cancer proliferation in in vitro cell lines, androgen dependent LNCap (the strongest effect) and androgen independent PC3 and DU145 in accordance with the abundant expression of GLP-1R in LNCap cells. Indeed, this anti-proliferative effect was abolished by GLP-1R antagonist or GLP-1R knockdown. Ex-4 attenuated in vivo prostate cancer growth induced by transplantation of LNCap cells into athymic mice, significantly reducing tumor expression of P504S, Ki67 and phosphorylated ERK-MAPK mediated by the cAMP-PKA pathway [87].

Some authors, using the GLP-1RA Ex-4, observed, via the PI3K/AKT/mTOR pathway, a reduction in the resistance of prostate cancer cells to the androgen-receptor inhibitor enzalutamide with a greater effect observed in advanced cancers. The increased chemosensitivity of cancer cells could be due to indirect inhibition of the tumor migration, invasion and growth [88].

In human prostate cancer cells (PC3 and LNCap), Ex-4 is able to promote the anti-proliferative effects of radiation via the activation of AMPK phosphorylation and the subsequent inhibition of phosphorylated mTOR, cyclin B and p34cdc2 protein kinase activation [89].

By binding with GLP-1R, liraglutide also significantly inhibited cell proliferation and induced cell apoptosis in LNCap prostate cancer cell lines by regulating the P38 pathway [90]. Liraglutide, in combination with Docetaxel, a first-line chemotherapy agent in metastatic castration-resistant prostate carcinoma, caused cell cycle arrest in LNCaP and induced apoptosis via the ERK/MAPK and AKT/PI3K pathways, thus suggesting new therapeutic strategies for lower therapeutic doses of docetaxel and, therefore, its resistance and toxicity [91]. Forced overexpression of GLP-1R using a lentiviral vector (ALVA-41-GLP-1R cells) was able to attenuate prostate cancer cell proliferation by inhibiting cell cycle progression in vitro and in vivo; the authors concluded that GLP-1R activation could be a potential therapy for prostate cancer [92].

### 4.3. GLP-1 Receptor Agonists and Pancreatic Cancer

Recent meta-analyses and reviews evaluating the association between GLP1-RAs and pancreatic cancer have found no significant associations in the use of incretin-based therapies and increased risk of pancreatic cancer [93,94]. On the contrary, it should be considered that obesity could be a risk factor for the development of pancreatic carcinoma [95]. Indeed, preclinical experimental data have shown beneficial effects of GLP1-RAs on pancreatic cancer cell lines. Zhao et al. [96], in human pancreatic tumor tissues, showed lower levels or lack of expression of GLP-1R than in pancreatic tissues adjacent to the tumor. GLP-1R negative expression was more frequent in advanced tumors and was associated with a poor prognosis. GLP-1R activation by liraglutide produces an antitumor effect on human pancreatic cancer cells through inhibition of the PI3K/AKT pathway, suggesting beneficial effects of GLP-1-based therapies in this tumor [96]. GLP-1RAs could also activate cAMP, subsequently inhibiting AKT and ERK1/2 signaling pathways and causing apoptosis and inhibition of proliferation in a human pancreatic cancer cell line in vitro and reducing xenograft pancreatic tumor growth in vivo, contributing to address the long-term safety issues of GLP-1-based therapies [97]. In addition, liraglutide, which regulates the NF-kB signaling pathway and the downstream ATP-binding cassette subfamily G member 2 (ABCG2), showed significant antiproliferative and pro-apoptotic effects in gemcitabine-resistant human pancreatic cancer cells resistant to various drugs, increasing the chemosensitivity in both in vitro and in vivo experiments [98].

### 4.4. GLP-1 Receptor Agonists and Gynecological Cancers

On gynecological cancers, GLP-1RAs exert an inhibitory effect on cervical cancer growth in T2DM; treatment with the antidiabetic GLP1-RAs attenuated hyperglycemia might promote cancer growth by increasing proteasome alpha subunit 2 (PSMA2) expression [99].

Upregulation of GLP1-R is able to block the growth of endometrial cancer cells (Ishikawa and RL95-2) by activating the cAMP/PKA signaling pathway [100]. Furthermore, liraglutide was able to suppress the progression of endometrial cancer in human Ishikawa endometrial cancer cells. GLP-1R expression is associated with estrogen and progesterone receptors positive status. Higher GLP-1R expression may be associated with a better prognosis in patients with endometrial cancer, and the use of liraglutide to target autophagy in endometrial cancer cells may be a potential new treatment for endometrial cancer [101].

GLP-1R is expressed in both human ovarian cancer tissues and cell lines. GLP1-R activation by Ex-4 is able to produce antitumor effects in human ovarian cancer cells by reducing growth, migration and invasion and promoting apoptosis through inhibition of the PI3K/Akt extension. GLP-1 activation is also able to attenuate tumor formation by ovarian cancer cells in vivo in nude mice [102].

It is known that tumor metastases are facilitated by the remodeling of the extracellular matrix at the tumor site and also through an alteration of the balance between metalloproteinases (MMPs), a family of proteolytic enzymes that contribute to the breakdown of extracellular matrix components, and the tissue inhibitors of metalloproteinases (TIMPs), promoting tumor growth, invasion and metastasis. MMP production also enhances the angiogenic response by VEGF expression potentially affecting the metastatic potential of cancer cells. In SKOV-3 and CAOV-3 human ovarian cancer cells, GLP1-RA (Ex-4) was able to reduce the expression of the key metalloproteinases MMP-2 and MMP-9, to modulate their inhibitors TIMP-1 and TIMP-2, to inhibit migration and induce apoptosis by reducing the production of adhesion molecules and inhibiting apoptosis in TNF-α-stimulated endothelial cells [103,104].

### 4.5. GLP-1 Receptor Agonists and Colorectal Cancer

In CT26 colon cancer cells, GLP-1R activation reduces cell growth and survival, increasing intracellular cAMP levels and inhibiting the activity of glycogen synthase kinase 3 and ERK1/2, a member of the mitogen activated protein kinase family; moreover, GLP-1R activation augmented apoptosis is induced by irinotecan, a topoisomerase I inhibitor [105].

By inhibiting the PI3K/Akt/mTOR signaling pathway, liraglutide could block the cell cycle, reducing cell proliferation, migration and invasion and promoting apoptosis in colorectal cancer cell lines. Liraglutide reduced cyclin D1, an important protein in the cell cycle that regulates the transition from G1 to S phase in the cell proliferation cycle, and MMP-11, one of the matrix metalloproteinases, which is the main enzyme that causes degradation of the basement membrane and of the extracellular matrix [106].

GLP-1R also appears to play a role in the inhibition of glioma cell survival, migration, proliferation and invasion in a GLP-1R/SIRT3 pathway-dependent manner [107].

## 5. Conclusions

It is now widely known that overweight and obesity are risk factors for cancer development and prognosis. Therefore, the prevention and treatment of obesity appears to be of increasing value as a treatment of a modifiable factor causing cancer. Extensive ongoing research about the effects of obesity treatment, especially focused on GLP1-RAs, on occurrence and development of cancer have highlighted the increased potential for their use in the clinical setting of cancer treatment, first reducing obesity/ABCD and, second, influencing cancer cells by altering proliferation, apoptosis, extracellular matrix remodeling and the response to chemotherapy. Further studies, including large, long term clinical trials are needed to properly evaluate the relationship between anti-obesity treatment and cancer risk and prognosis, and the long-term efficacy of these agents on weight control versus lifestyle or diet changes. These studies require strong collaboration between oncologists and experts in the treatment of obesity.

## Data Availability

Not applicable.

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
