# Peer review of "Novel Treatments for Obesity: Implications for Cancer Prevention and Treatment"

_nutrients, 2023, doi:10.3390/nu15173737_

Round 1

Reviewer 1 Report

The author reviewed the correlations between weight loss in obesity and cancer, which echoes the obesity pandemic around the world. The implication of Metformin and GLP1-receptor agonists in cancer are also comprehensively described. This critical review provides new insights into cancer prevention and treatment.

Minor comments

1.     Since Leptin is well known to be involved in obesity treatment, it would be nice if Leptin’s implication in cancer was included.

2.     Brown fat mediated global metabolism alteration was reported to suppress tumor growth (PMID: 35922508), this cutting-edge research will make the manuscript more attractive if it was introduced in the manuscript. 

3.     It seems there is a grammatical error in Line 181.

Author Response

Thank you so much for your kindly review:

1- Thank you for your suggestion. We added a sentence about the role of leptin in cancer on page 2.

2- We added some brief info about the role of BAT in tumorigenesis on page 2.

3- The error was corrected.

Best regards

Reviewer 2 Report

Thank you for your review article to this journal.

In this article, you have demonstrated the importance of obesity treatment and mechanism and partial research results in cancer treatments, esp, GLP1A and various cancers.

Through reading this article, I feel it seems to be of great help researchers and therapists by understanding mechanisms of obesity treatment on cancer managements.

I hope you will upgrade the research results in future. 

Author Response

Dear Reviewer,

thank you very much. We are delighted you liked the manuscript.

Best regards